# Uncovering the Effects of the Cultivation Condition on Different Forms of Peptaibol's Emericellipsins Production from an Alkaliphilic Fungus, *Emericellopsis alkalina*

Anastasia E. Kuvarina [1,*], Maxim A. Sukonnikov [1,2], Alla V. Timofeeva [3], Marina V. Serebryakova [3], Ludmila A. Baratova [3], Mikhail N. Buzurnyuk [2], Alexander V. Golyshkin [1], Marina L. Georgieva [1,4] and Vera S. Sadykova [1,*]

[1]  Department of Microbiology, Gause Institute of New Antibiotics, St. Bolshaya Pirogovskaya 11, 119021 Moscow, Russia; sukonnikoff.maxim@yandex.ru (M.A.S.); i-marina@yandex.ru (M.L.G.)
[2]  Faculty of Chemistry, Lomonosov Moscow State University, 119991 Moscow, Russia; m_buzur@mail.ru
[3]  Belozersky Institute of Physico-Chemical Biology, Lomonosov Moscow State University, 119991 Moscow, Russia; 2.04.cochon@gmail.com (A.V.T.); kukarino@yandex.ru (L.A.B.)
[4]  Faculty of Biology, Lomonosov Moscow State University, 1-12 Leninskie Gory, 119234 Moscow, Russia
*   Correspondence: nastena.lysenko@mail.ru (A.E.K.); sadykova_09@mail.ru (V.S.S.)

**Abstract:** Peptaibols (Paib) are a class of biologically active peptides isolated from fungi and molds, which have attracted the attention of medicinal chemists due to their widely ranging pharmacological properties, including their antimicrobial activity. In the present study, we investigated the effects of various pH levels and cultivation conditions on peptaibol complex emericellipsins A-E (EmiA-E), produced by the alkaliphilic fungus *Emericellopsis alkalina*. Paib production has been studied in flasks and bioreactors for different pH values ranging from 7 to 11. The study of morphological features based on light and scanning electron microscopy has revealed differences between fungi grown at different pH values and cultivation conditions. Emericellipsins have been purified, sequenced, and identified by mass spectrometry. We have found that an alkaline pH of 10 could promote emericellipsins' biosynthesis most effectively during stationary fermentation on the 14th day of cultivation.

**Keywords:** antimicrobial peptide; alkaliphilic fungi; *Emericellopsis alkalina*; emericellipsins A-E; antifungal; antimicrobial resistance

## 1. Introduction

Growing concerns about antimicrobial resistance in human pathogens have led to the research of alternative strategies to prevent them. As a result, new bioactive compounds, such as antimicrobial peptides, have been discovered [1–4]. These compounds can be exploited to control multidrug-resistant pathogens [5–7]. Among the antimicrobial peptides, the class of non-ribosomal peptides, known as peptaibols (Paib), may be noted for its broad range of biotechnology applications. Paib are composed of chains of 5 to 20 amino acid residues, with high percentages of lipophilicity [6–10]. In particular, they are caused by the presence of non-amino acids, such as α-aminoisobutyrate (Aib), isovaline (Iva), and isoleucine (Ile). In addition, Paib also contain the acetylated N-terminal and the C-terminal in the form of an amino alcohol [11–13]. Paib have great potential to produce new antimicrobial drugs, as they work by destabilizing the lipid bilayer, which decreases the capability of developing resistance mechanisms. The activity of Paib against nematodes, pathogenic bacteria, and fungi, as well as viruses, has been announced [14].

It is known that this group of non-ribosomal peptides is synthesized by micromycetes from the genera *Trichoderma*, *Acremonium*, *Emericellopsis*, and a few others. A producer normally synthesizes a complex of peptaibols, which are structurally homologous compounds that vary in their placement in the peptid chain by one or several amino acids, which



also determines differences in bioactivity [15–22]. The discovery of Paib in micromycetes that inhabit cold and saline soils, sea depths, and other extreme environments expands opportunities for generating new antimicrobial drugs [23–25]. Specifically, the alkaliphilic species *Emericellopsis alkalina* from soda soils has recently been described as an efficient Paib producer. Previously, we indicated that the complex contains at least five peptides with a single amino acid substitution produced by *E. alkalina*, with broad antimicrobial activities [26–30]. The main compound of this complex, emericellipsin A (EmiA), had considerable antifungal activity against the clinical isolates of pathogenic fungi with multiple drug resistance [26–30]. EmiA has approved high antifungal activity for azole-resistant clinical isolates of *Aspergillus* species, comparable to amphotericin B [29].

In this study, emericellipsins' profiles and total amounts of Paib were analyzed during different cultivation processes. Afterwards, mass spectrometry techniques were applied for the identification and sequencing of Paib. Furthermore, the effects of potential relationships between the cultivation conditions on fungi micromorphological structures and peptaibols' production were carried out.

## 2. Materials and Methods

### 2.1. Cultivation of the Fungi and Extraction of Emericellipsins

The quantitative peptaibol profiles and structural diversity of emericellipsins in culture broth were assessed using the type strain of the mycelial fungus *Emericellopsis alkalina* E101 (VKM F4108; CBS 127350), from the collection of Fungi from Extreme Environments, Department of Mycology and Algology, Faculty of Biology, Moscow State University (Russia). This strain was isolated from a soil sample taken on the coast of the soda lake Tanatar (Altai Krai, Russia) [31]. Previously, it was shown that the strain *E. alkalina* E101 was most productive with respect to the yield of the major compound EmiA [29,31].

The producer strain *E. alkalina* E101 was grown in a specialized liquid alkaline medium previously selected, at 28 °C for one week. Then, a filtered spore suspension ($1 \times 10^6$ spores mL$^{-1}$) was arranged as an inoculum by cytometry. The growth media consisted of a carbon source, according to the previous protocol [29]. The effects of pH on the growth rate and EmiA were evaluated in triplicate, with several based on phosphate and carbonate buffers (pH 7, 9, 10 ($\pm$0.2 for each point)). The contents of the alkaline media and buffers' composition followed the work of Grum–Grzhimaylo et al. [31].

The study of fungal morphology was investigated using a light microscope (LM) and a scanning electron microscope (SEM). LM observations were performed on a Leica DM2500 microscope equipped with a DFC 495 camera. Samples for SEM were fixed with formaldehyde (approximately 4% final concentration), followed by washing and dehydration steps (20 min each) with a series of ethanol (30%, 50%, 70%, and 96% concentrations) and acetone before drying at the critical point in $CO_2$, carbon, and metal coating. SEM observations were performed using Camscan-S2 and JEOL, JSM-6380LA, with an accelerating voltage of 20 kV, in SEI mode. Images were acquired and elaborated using MicroCapture software.

For the production of EmiA, fungi were cultivated in 750 mL Erlenmeyer flasks in an Innova 40R shaker–incubator (Eppendorf New Brunswick, NJ, United States) under stationary rotation conditions for 7–21 days, and in a bioreactor for 10 days. The culture fluid (CF) was separated by filtration through membrane filters on a Seitz funnel under a vacuum. Every three days after the seventh day of cultivation, samples were taken for measuring biomass production (dry weight) and the production of peptides by mass spectrometry. CF was separated by filtration through membrane filters in a Seitz filter funnel under a vacuum. The CL was extracted 3 times with ethyl acetate (EtAOc) at a ratio of 5:1. The resulting extracts were evaporated under a vacuum in a Rotavapor-RBüchi (Buchi, Flawil, Switzerland) at 42 °C to dryness, and the remainder was dissolved in aqueous 50% ethanol to obtain alcohol concentrates.

*2.2. MALDI-TOF MS Analysis*

All samples had a volume of 20 mL. They were dried in 2.0 mL plastic tubes at a temperature of 46 °C (Fisher scientific, Waltham, MA, USA), applying SpeedVac Vacuum Concentrator (Thermo Fisher, Waltham, MA, USA) in a few repeats with EtOH (Sigma-Aldrich, St. Louis, MO, USA) additives. Then, the solid–liquid extraction was performed with 4 portions of EtOAc (Sigma-Aldrich, St. Louis, MO, USA) with a volume of 1 mL. Within the process of extraction, after every 1 mL of EtOAc had been added, the probes were sonicated with ultrasound bath equipment (Thermo Fisher, Waltham, MA, USA) to dissolve all solid pieces. Then, the supernatant was centrifuged at 15,000 rpm for 15 min, applying a microcentrifuge (Thermo Fisher, Waltham, MA, USA), and then it was isolated to another plastic tube. Finally, it was entirely dried by extracting the liquid components, applying a SpeedVac Vacuum Concentrator (Thermo Fisher, Waltham, MA, USA) at 40 °C, and solids were then diluted with 100 μL of 50% EtOH ($H_2O$, *v:v*).

All the culture broth samples had a volume in the range of 130 to 150 mL. These solutions were extracted (l:l) with EtOAc (Sigma-Aldrich, St. Louis, MO, USA)in two steps: firstly, in the proportion 1:2 (cultural medium:EtOAc), and then 1:1 (cultural medium:EtOAc). Extracts from every step for one sample had been combined in a flask and then were entirely dried with a rotary evaporator (Thermo Fisher, Waltham, MA, USA) at 45 °C. The next step was the dissolution of dried solids with 4 mL of 50% EtOH ($H_2O$, *v:v*) in two portions of 2 mL. After it had been dried with the SpeedVac Vacuum Concentrator (Thermo Fisher, Waltham, MA, USA) application at 40 °C, it was then diluted with 100 μL of 50% EtOH ($H_2O$, *v:v*)

For MALDI-TOF MS analysis, 0.3 μL of EtOH (50% in MQ) solution or MeCN-$H_2O$ fraction (collected during HPLC separation) samples, and 0.5 μL of 2,5-dihydroxybenzoic acid (Sigma-Aldrich, Darmstadt, Germany) solution in 20% MeCN + 79.5% water (MQ) + 0.5% TFA (HPLC, Sigma-Aldrich, St. Louis, MO, USA) in a concentration of 20 mg/mL, were mixed on a target of the spectrometer. The specter recording and MS analysis were carried out with utilization of a MALDI-TOF MS spectrometer UltrafleXtreme BrukerDaltonics (BrukerDaltonics, Bremen, Germany), equipped with a UV-laser (Nd) in the positive ion registration mode, utilizing a reflectron. The accuracy of mass detection was about 1 Da. All the calculations and the plots' creation were performed with the Microsoft Excel 2016 software application.

## 3. Results

*3.1. Effects of the Initial pH and Cultivation for Growth, Sporulation, and Emericellipsins Complex Production*

Previous studies have demonstrated that the main compound EmiA's secretion increased at an alkaline pH compared to a neutral pH [26–30]. This result reflects both the effect of the initial pH and the type of cultivation on the mycelial growth and the production of antifungal Paib. To obtain a global view of the effects of various cultivation conditions on the production of different forms of emericellipsins complex, we compared strain E101 cultured at pH 7, 9, and 10. The influence of the pH value and cultivation method on the micromorphology of *E. alkalina* was analyzed (Figure 1). The formation of emericellipsins complex and biomass by the producer was studied at neutral and alkaline initial pH values under stationary (Figure 1d) and submerged cultivation (Figure 1c), as well as in a bioreactor (Figure 1a,b), from 7 to 21 days.

It was found that *E. alkalina* E101 was able to form both conidial sporulation and the perfect stage (fruiting bodies) when cultivated on a neutral agar medium (MA, malt agar, pH 6.5) (Supplementary Figure S1). When cultivating on media with an alkaline pH, for both agar and liquid, only conidial sporulation was formed.

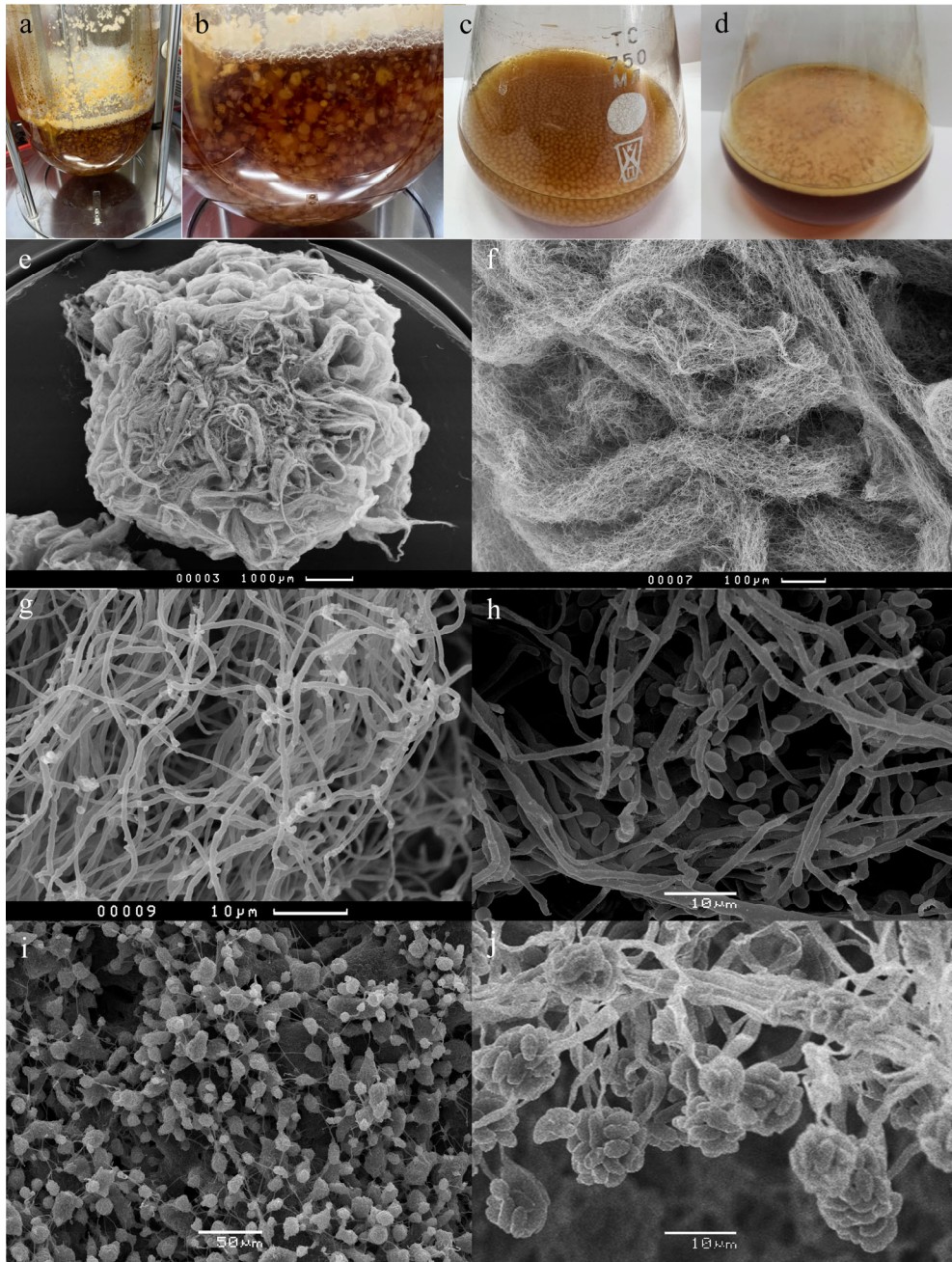

**Figure 1.** Cultivation conditions and micromorphology (SEM) of *E. alkalina* E101: (**a**,**b**,**e**–**g**) bioreactor, (**c**,**h**) submerged condition, and (**d**,**i**,**j**) stationary condition.

The *E. alkalina* E101 strain had a different ability to form spores (conidia) depending on the cultivation method. During stationary cultivation, the maximum spore formation was noted (Figure 1i,j). Numerous phialides were noted both on single hyphae and on bundles of mycelium. Under the conditions of submerged fermentation, *E. alkalina* E101 was able to produce conidia and phialides typical of this species (Figure 1h), but much less abundantly than under stationary conditions (Figure 1i,j). The bioreactor provides an optimal environment for the growth of the strain and is favorable for the accumulation of biomass (Figure 1e–g). At the same time, a complete inability in *E. alkalina* E101 to form phialides and conidia was revealed. It was further observed that Paib were not formed during vegetative growth in the bioreactor (data not shown).

The variations in the biomass production and the total amount of peptaibols are shown in Figure 2. It was observed that the biomass production was much lower in stationary cultivation than that in submerged cultivation in all pH variants. The neutral pH served as an indication to the cell that conditions were appropriate for growth within the phase with exponential growth of fungus. The maximum biomass growth was seen at pH 7, regardless of the type of cultivation.

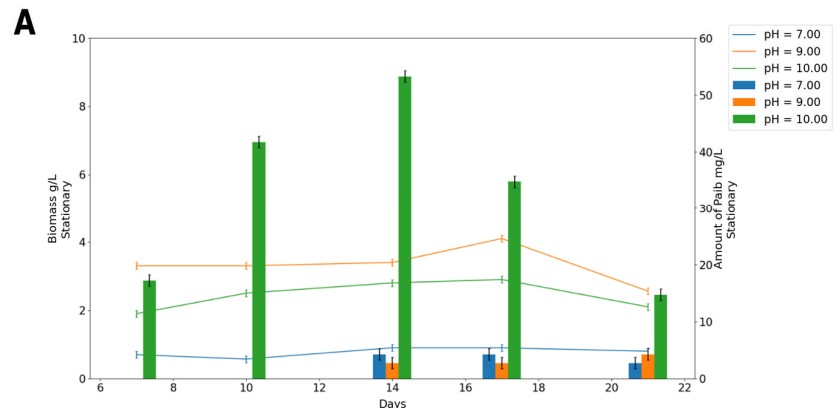

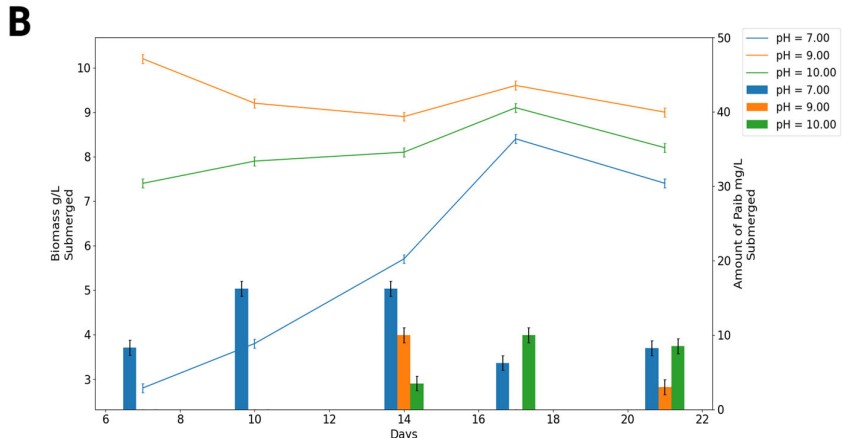

**Figure 2.** Production of Paib (bars) and biomass (lines) according to the different cultivation conditions: (**A**) stationary condition and (**B**) submerged condition.

The alkaline culture medium with pH 10 significantly increased the production of emericellipsins during the stationary cultivation (from 9 to 14 days), while biomass generation was reduced during 14 to 21 days. The highest biomass amount was recorded on day 14 of growth at all initial pH values in the medium; however, the maximum peptaibols content was observed at alkaline pH values (10). There was no detected Paib production in the initial cultivation period (lower than 5 days) (Figure 2).

The maximum rate of mycelial growth was observed under the submerged cultivation in the flask and the bioreactor compared the stationary cultivation. An increase in the biomass amount was detected, followed by the stationary phase, after the seventeenth day of cultivation in all types of cultivation and at all pH values. On the contrary, the maximum amount of Paib complex was seen at a neutral pH value, but much lower than that in all stationary variants.

### 3.2. EmiA and Homologous Forms' Identification

After the investigation of all pH values and variants of cultivation samples, their Paib compounds were apparent. No forms of emericellipsins were detected in the bioreactor until ten days of cultivation.

In order to find the homologous forms, all the masses registered within MS spectrometry from different samples were collected together to form one big dataset, including all the resulting m/z and intensity values. At the same time, sets of theoretical masses related to both ionized EmiA forms, such as [EmiA + H]$^+$ and [EmiA + Na]$^+$, were formed. These theoretical masses were obtained using the following formulas: $M = 1049.7 + 18y + 14x + 1$ (digit 1 relates to the mass of H,H-ionized form), and $M = 1049.7 + 18x + 14y + 23$ (digit 23 relates to the mass of Na,Na-ionized form). Both $x$ and $y$ variables are digits, which vary from –5 to 5 and reflect the number of –CH$_2$– or –CH$_3$ radicals and water molecules, respectively. Particles –CH$_2$– or –CH$_3$ and H$_2$O, with masses of 14 Da and 18 Da, respectively, are probable molecular differences between both of the molecular forms of EmiA homologues. Dealing with x and y values out of the exhibited range will extremely decrease the prediction probability due to the extremely increasing values of mass and the variety of probable molecular combinations, including C, H, O, and N atoms. Thus, experimental mass values were crossed with the theoretical masses to find all the overlaps, indicating the potential homologues to be selected.

The scheme of the research is presented in Figure 3, and the process of selection is exhibited in Supplementary Figures S2 and S3.

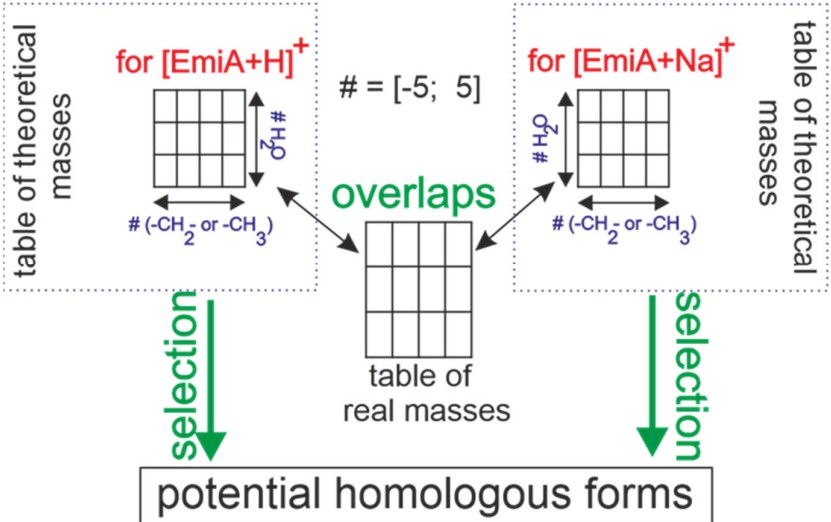

**Figure 3.** Schematic presentation of homologues' selection. # Is a digit parameter changing in a range of –5 to 5.

Masses of 1022.7, 1036.8, 1044.7, 1050.6, 1052.7, 1058.5, and 1072.6, as homologous forms, were selected within the observation. Several masses, such as 1050.6, 1036.8, and 1052.7, correlated with the forms EmiA, EmiB, and EmiD annotated in [29]. On the other hand, the masses 1066.76 and 1079.75 were also annotated but have not been registered within the current screening. The form with the mass of 1032.7 (dEmiA) has also not be detected.

All the results of the homologous form screening were collected in one plot, as exhibited in Figure 4.

The obtained dash plot reflects the molecular relationships between all the selected homologous forms, where the masses colored in red are considered to correlate with the [M + H]+ ionization type, and those colored in green with the [M + Na]+ type.

It is clear that the masses of 1022.7, 1036.8, and 1050.8 (EmiA) formed a set, where the difference is expressed in –CH$_2$– or CH$_3$– particles between one form and the next with a higher mass. The same dependence was achieved with the set of masses of 1044.7, 1058.5, and 1072.6. Moreover, the masses of 1022.7, 1036.8, and 1050.6 overlapped with the masses 1044.7, 1058, and 1072.6, so the latter masses are likely to be [M + Na]+ forms for the first set of masses ([M + H]+), correspondingly.

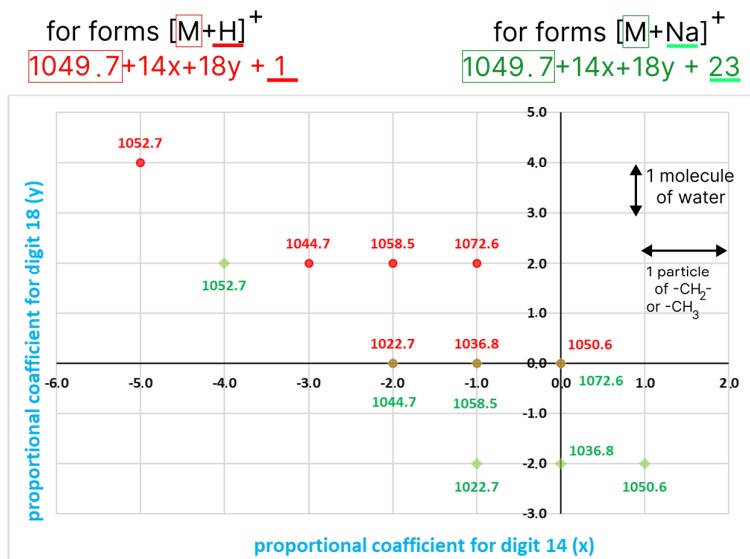

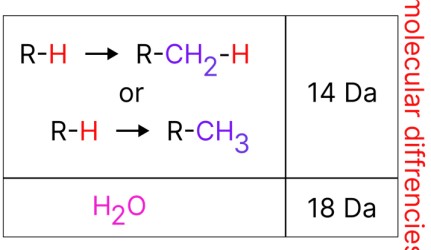

**Figure 4.** The plot of the selection results. The *x*-axis reflects molecular differences in–CH$_2$– or CH$_3$– particles, and the *y*-axis reflects molecular differences in water molecules. The plus direction indicates the increasing number of substituents, and the minus direction indicates decreasing. Points colored in red relate to the [M + H]+ type, and those in green to the [M + Na]+ type. In the right corner, the information about types of substitution and their mass differences are depicted.

### 3.3. Exploring Emericellipsins Forms' Production in Different Conditions

In order to estimate Paib's sensitivity to the cultivation condition changes, we analyzed m/z and intensity values' dependencies from values of different attributes. Within the experiment, we observed that there were several forms with only one mass, so every point of the same mass characterized only one sample. Thus, we created dot plots (Supplementary Figures S2 and S3), where we calculated the number of dots (such as the number of samples with this mass) with the same m/z value of any Paib form (Supplementary Figure S3) and the number of dots with an intensity value higher than 10,000 (Supplementary Figure S4). Figure 5 summarizes the effects of pH values and the various cultivation conditions.

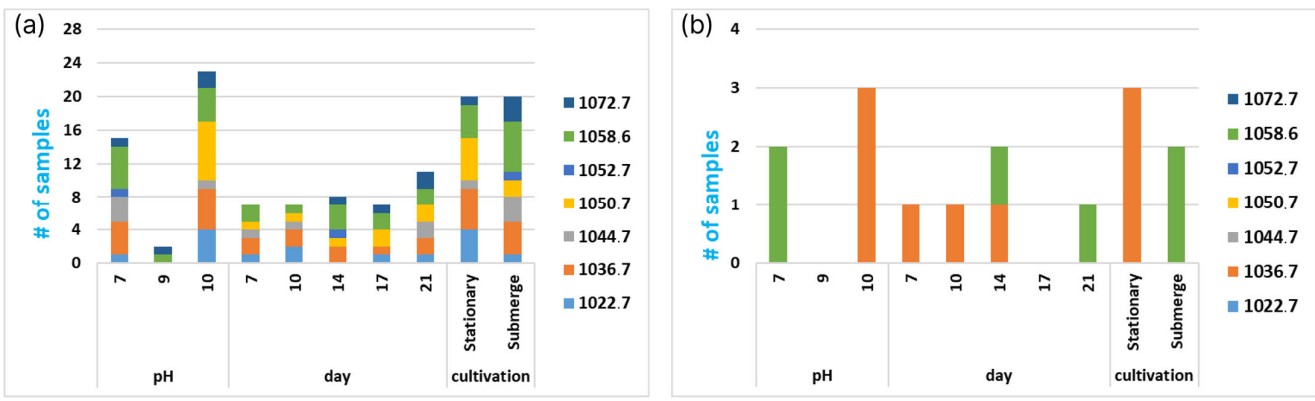

**Figure 5.** Mass distribution by local attributes. (**a**) The histogram of the number of samples depending on different local attributes. The color key is presented in the right corner. (**b**) The histogram of the number of samples with an intensity higher than 10,000 c.u., depending on different local attributes. The color key is presented in the right corner.

It was detected that the most suitable pH for Paib production was 10; pH 7 was also appropriate, but the total score of Paib was lower (Figure 5a). Production was minimal with

pH 9. EmiA can only be produced at pH 10. The major forms at all conditions were 1036.7 and 1058.6. Antifungal activity against the multidrug-resistant *Aspergillus* spp. isolate has been previously shown for the forms of emericellipsins with masses of 1036.7 Da (EmiB) and 1050.8 Da (EmiA) [28,29].

The form 1050.8 (EmiA) had a pH optimum of 10 (at pH 7 and 9, it existed in trace amounts). Both cultivation types at 1050.8 were produced, but the maximum content related to the stationary type of cultivation (Figure 5b). There were two molecular forms with a high level of production, 1036.7 and 1058.6 Da, regardless of the type of cultivation. The form 1036.7 had an optimum pH of 7 and 10. It existed within all periods of cultivation, with a maximum at day 14 and a minimum at day 17. The form 1022.7 existed at pH 7 and 10 (with maximum content at pH 10). The maximal content of 1022.7 was at day 7, and then it extremely decreased after day 10. Both types of cultivation were suitable for the form 1022.7, but the maximum content was achieved with stationary cultivation. The form 1052.7 had trace amounts within the whole experiment—it appeared once at pH 7, on day 14, with the application of submerged cultivation.

### 3.4. Exploring the Global Forms' Distribution by All the Attributes

We can observe the principal difference between the two types of cultivation in Figure 6a, and the accumulation type histogram in Figure 6b. Thus, peptaibol's production only took place at pH 10 with stationary cultivation, and in general at pH 7 with submerged cultivation application. However, on day 14, we observed a small amount of peptaibol's production at pH 9 and on day 17 at pH 10. This probably occurred due to the secondary metabolites' consumption by fungi, which changed the character of non-ribosomal synthesis.

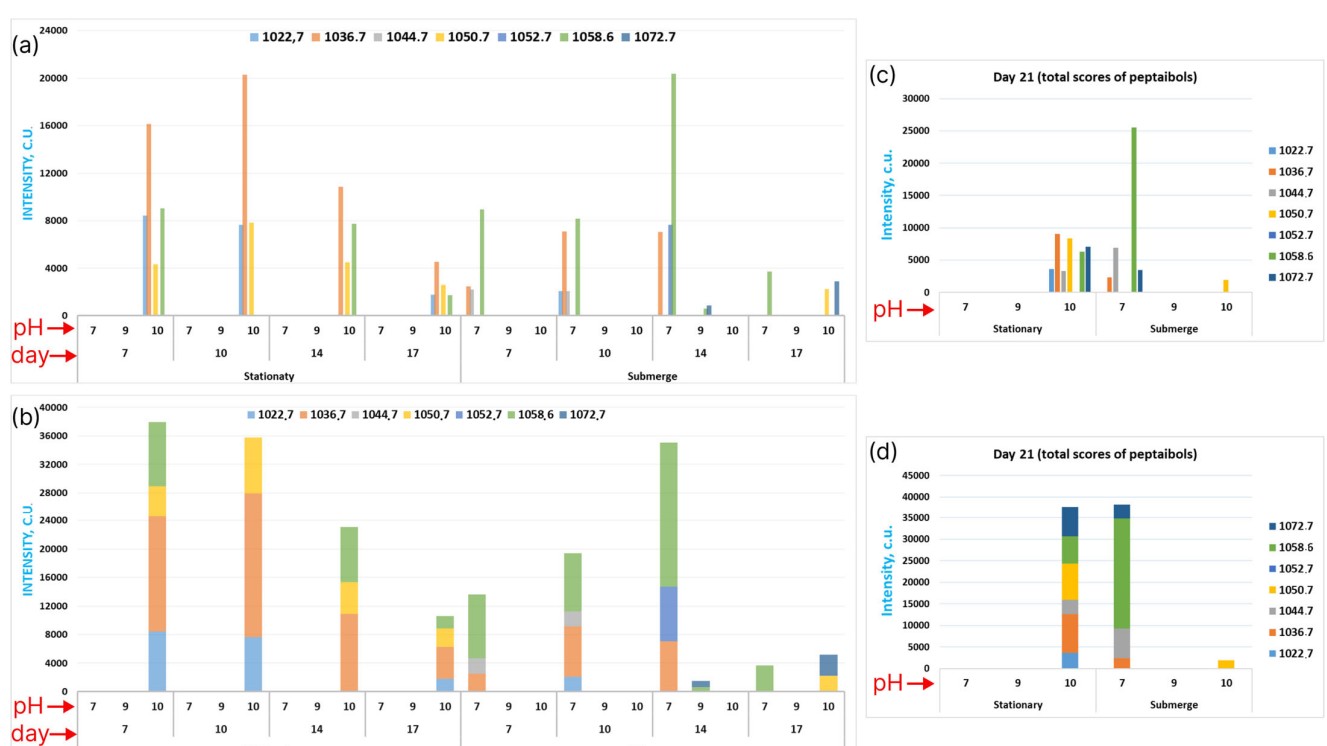

**Figure 6.** Mass distribution through all the attributes. (**a**) The histogram of intensities depending on all the attributes. The color key is presented at the top. (**b**) The same as (**a**), in the form of the accumulated type of histogram. (**c**) The histogram of intensities depending on all the attributes at day 21. (**d**) The same as (**c**), in the form of the accumulated type of histogram.

In addition, it is interesting to point out the dynamics of peptaibol's formation. For stationary cultivation, a steady decrease of peptaibol's production was seen from day 14 to day 17, while the reverse situation was seen for the submerged type of cultivation, whereby

we observed a steady increase of peptaibols toward day 14, but then an extreme decrease occurred. The decrease of the amount of peptaibols toward day 21 of cultivation can be explained by the metabolites' consumption by fungi. Other phenomena depend on the increasing local nutrition and [OH$^-$] concentrations and the surface area of fungi due to the constant stirring.

Finally, it is shown in Figure 6c,d that stationary cultivation yielded a high distribution of Paib, and the amount of EmiA was much higher than in another cases, while submerged cultivation yielded a lower distribution, but EmiA production was minimal. Despite these results, our understanding of the regulatory mechanisms of different forms of emericellipsins' synthesis in *E. alkalina* is limited, which hampers the study of efficient cultivation conditions for EmiA production.

## 4. Discussion

Fungal morphology is an essential criterion that determines the mycelial growth rate and the properties of the fermentation broth in the producers' cultures [32–35]. In the present work, microscopic observations of morphological features of *E. alkalina* cultured in different conditions and at different pH values were performed. The amounts of spores proved to be dependent on the cultivation type. To date, the micro-characteristics of the conidia and phialides structures of this fungus have been reported from stationary and submerged cultivation in flasks. As the results have shown, the producer *E. alkalina* E101 strain could only form maximum conidia in stationary variants of cultivation, which were characterized by the lowest biomass accumulation.

The alkaline medium significantly increased the production of Paib, while biomass generation was reduced. The production of Paib only started on the seventh day of cultivation, showing a maximum in alkaline medium, where numerous conidia and phialides were observed. A study of the micromorphology of *E. alkalina* confirmed that sporulation under stationary conditions at alkaline pH is excellent, in contrast to submerged cultivation. Some authors previously demonstrated that sporulation plays a role in the synthesis of Paib by *Trichoderma* spp. [20,36]. Komon-Zelazowska et al. previously demonstrated that sporulation plays a role in the synthesis of Paib by *Hypocrea/Trichoderma* spp. [37,38]. Thus, we presumed that the emericellipsins synthesis would reach its optimum under formulation of the film on a medium surface and would depend on the production of numerous conidia. No Paib were formed during vegetative growth by surface-grown cultures of *Hypocrea/Trichoderma atroviridis*, but a micro-heterogenous mixture of atroviridins accumulated when the colonies commenced sporulation. This correlation between sporulation and atroviridin formation was detected to be independent of the pathway inducing sporulation (i.e., light, mechanical injury, and carbon starvation, respectively) [37,39]. Presently, these results show that the best biomass accumulation is not always correlated with a greater production of antibiotics.

Moreover, intraspecific changes may modify the production of these peptides, depending on the conditions in which each fungus grows [4,40]. The differences in Paib production due to varied growing conditions have been observed by many groups of researchers. Thus, the addition of glucose to the medium led to an increase in biomass accumulation but lowered the Paib production for *Trichoderma* species. The decrease in Paib yield could be interpreted by the presence of glucose sensor homologs and transcriptional regulators that negatively regulate genes encoding for NRPS when saturated with glucose [40,41].

Putative methyl transferase plays a significant role in regulating Paib production. Deletion of Tllae1 resulted in a negligible impact on conidial production and decreased the production of Paib. The results demonstrated that this defect occurred at the transcriptional level of the two synthetases-encoding genes, tlx1 and tlx2, which are responsible for Paib production. In contrast, constitutive expression of Tllae1 in *T. longibrachiatum* SMF2 led to two-fold increased Paib production. The production of conidia in ΔTlstp1 was significantly reduced by 50% compared to the WT strain, indicating that the removal of Tlstp1 also led to a marked defect in conidia formation. The defect in both vegetative growth and

sporulation was almost completely restored when ΔTlstp1 was complemented with an expression cassette for in situ expression of Tlstp1, further indicating that TlSTP1 plays an important role in the growth and conidiation of *T. longibrachiatum* SMF2 [42].

Tamandegani et al. announced that Paib production significantly increased upon in vitro interaction with *Fusarium oxysporum* [14]. The addition of Aib increased Paib production due to its immediate availability in the culture medium. The amino acid Aib increased the synthesis of Paib of *T. asperellum* when added to the culture medium during the stationary phase [4].

The microheterogeneity due to the flexibility of some NRPS modules allows for obtaining many trichotoxin isoforms in *T. asperellum*. These isoforms can vary from one another in a single unit of mass, as occurs between trichotoxins 1703A and A-40 [39,43]. Intraspecific differences can vary the production of these isoforms, depending on the environment in which each fungus develops [4].

Alkaliphilic and alkali-tolerant fungi can exhibit differences in tolerance to the medium that simulates the conditions of their natural habitation [44,45]. Therefore, it would be ideal to have a quantitative model allowing the prediction of cultivation conditions for Paib profiles' synthesis and enhancing the amount of Paib. For the cultivation conditions which were used here, it can be seen that the global physiological regulation is very complex due to the diversity of fungi and the variety of metabolic pathways and controls [46,47]. In the present work, the pH also influenced the production of the emericellipsins complex, as well as the production of its different forms. Production of different emericellipsins forms was higher at pH 10; pH 7 was also appropriate, but the total score of Paib was lower. Production of Paib was minimal at pH 9. Along with the pH value, the type of cultivation may depress the synthesis of antibiotics [46]. Thus, the EmiA compound (1050 Da) can only be produced at pH 10. The major emericellipsins forms were identified as 1036.7 and 1058.6 Da. Masses of 1022.7, 1036.8, 1044.7, 1050.6, 1052.7, 1058.5, and 1072.6 were selected as potential homologue forms. Masses of 1022.7, 1036.8, and 1050.6 were likely to be the nearest forms, and the masses 1044.7, 1058.5, and 1072.6 were probably their Na-ionized forms. Mass 1052.7 seemed to be a further "relative". The 1036.7 form was a major component within the whole analysis.

A higher level of EmiA was observed under stationary conditions in all variants in this study, indicating that the Paib was better excreted to these conditions. The maximum content of EmiA was reached at pH 10, on day 14, applying the stationary type of cultivation. The experiment confirmed that EmiA was better produced under alkaline cultivation conditions, despite that *E. alkalina* strains were able to grow and develop within a wide range of pH values in the medium (7.0–11.0), with the growth optimum at pH 10. It was also indicated that stationary cultivation requires pH 10 and submerged cultivation requires pH 7, but the quantity and variety of peptaibols was dramatically low.

## 5. Conclusions

The results clearly support the previous conclusions of other researchers that optimized medium and culture conditions are the main control factors for the production of Paib compounds, as well as its various forms [4,12,14,48–51]. We previously found that some emericellipsins produced by *E. alkalina* exhibited promising antimicrobial activity against a range of Gram-positive bacterial and fungal pathogens. The results from this study suggest that the alkaliphilic fungus *E. alkalina* may represent an interesting source of known and new Paib and antimicrobial compounds of biotechnological interest. Future studies may incorporate the validation of the proposed treatment against other species and strains and the best use strategy for the products.

**Supplementary Materials:** The following supporting information can be downloaded at: https://www.mdpi.com/article/10.3390/fermentation9050422/s1, Figure S1: *Emericellopsis alkalina* E101 isolate: (**a**,**b**) 14-day-old colony on malt agar, forming both conidial sporulation and the perfect stage (fruiting bodies), (**b**) fruiting bodies with ascospores (LM), (**c**,**d**) 14-day-old colony on alkaline agar, only conidial sporulation is formed, (**d**) numerous conidial heads (SEM). Figure S2: Dot scattering plot for homologues selection with the form $[EmiA + H]^+$: (**a**) Zoom [870–970] Da, (**b**) Zoom [970–1080] Da. All the matches are indicated with red circles, and all the masses of interest with a red line. (**c**) Zoom [1090–1200] Da. All theoretical dots are colored in gray, and all experimental dots are colored in red. Figure S3: Dot scattering plot for homologues selection with the form $[EmiA + Na]^+$. (**a**) Zoom [900–1010] Da, (**b**) Zoom [1010–1110] Da. All the matches are indicated with red circles, and all the masses of interest with a red line. (**c**) Zoom [1120–1250] Da. All theoretical dots are colored in gray, and all experimental dots are colored in red. Figure S4: Sample selection with intensities higher than 10,000 c.u. by attributes of (**a**) pH, (**b**) day, and (**c**) cultivation type. All the dots reflecting the EmiA homologue that lies in the interval of interest are indicated with red circles.

**Author Contributions:** Conceptualization, V.S.S. and M.L.G.; methodology, M.L.G. and V.S.S.; software, A.E.K.; validation, A.E.K. and M.A.S.; formal analysis, A.E.K., M.A.S., M.L.G. and V.S.S.; investigation, A.E.K., M.A.S., A.V.T., M.V.S., L.A.B., M.N.B., A.V.G., M.L.G. and V.S.S.; data curation, A.E.K., M.A.S., M.L.G. and V.S.S.; writing—original draft preparation, A.E.K., M.A.S., A.V.T., M.V.S., L.A.B., M.N.B., A.V.G., M.L.G. and V.S.S.; writing—review and editing, A.E.K., M.L.G. and V.S.S.; visualization, V.S.S.; supervision, V.S.S.; project administration, V.S.S.; funding acquisition, A.E.K. All authors have read and agreed to the published version of the manuscript.

**Funding:** This study is supported by the Russian Science Foundation (Grant No. 21-75-00062, https://rscf.ru/en/project/21-75-00062/, accessed on 31 March 2023).

**Institutional Review Board Statement:** Not applicable.

**Data Availability Statement:** All sequence data are available in NCBI GenBank following the accession numbers in the manuscript.

**Acknowledgments:** SEM studies were carried out at the Shared Research Facility "Electron microscopy in life sciences" at Moscow State University (Unique Equipment "Three-dimensional electron microscopy and spectroscopy"). The authors are grateful to LaAnton A. Georgiev (Moscow State University) for helping with processing the images of fungi. The authors are grateful to Ruslan M. Sadykov for graphical data processing in Python.

**Conflicts of Interest:** The authors declare no conflict of interest.

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
