# Peer review of "Uncovering the Effects of the Cultivation Condition on Different Forms of Peptaibol’s Emericellipsins Production from an Alkaliphilic Fungus, Emericellopsis alkalina"

_fermentation, doi:10.3390/fermentation9050422_

Round 1
Reviewer 1 Report
This manuscript showed effects of various pH and cultivation condition on peptaibol complex emericellipsins A-E (EmiA-E) produced by alkaliphilic fungus Emericellopsis alkaline, and found that alkaline pH 10 could promote emericellipsins biosynthesis most effectively during stationary fermentation.
The part of Introduction was simple and did not present the research background clearly.
The 2.3, 2.4, 2.5 in Materials and Methods described similar types of experiments, so it was suggested to merge them together.
None of the data was statistically analyzed, and no standard deviation was indicated in the figure. And the quality of the pictures is poor.
Why only three groups of pH 7, 9, and 10 are selected? When pH exceeds 10, will the yield be higher, and I wonder about the biomass curve at Ph 9,please explain it.
There are a lot of spelling mistakes in the manuscript, such as abbreviations without full names, chemical formulas
Author Response
This manuscript showed effects of various pH and cultivation condition on peptaibol complex emericellipsins A-E (EmiA-E) produced by alkaliphilic fungus Emericellopsis alkaline, and found that alkaline pH 10 could promote emericellipsins biosynthesis most effectively during stationary fermentation.
Thank you very much for the evaluation of the manuscript. We are resubmitting our revised manuscript and detailed point-by-point responses. We appreciate your constructive comments!
The part of Introduction was simple and did not present the research background clearly.
We thank the reviewer for pointing this out. We have revised it. We added some extra information to the Introduction section.
The 2.3, 2.4, 2.5 in Materials and Methods described similar types of experiments, so it was suggested to merge them together.
Thank for the notification. The parts 2.3 and 2.4 have been merged into a single text.
None of the data was statistically analyzed, and no standard deviation was indicated in the figure. And the quality of the pictures is poor.
We added standard deviation in Figure 2A and Figure 2B
Why only three groups of pH 7, 9, and 10 are selected? When pH exceeds 10, will the yield be higher, and I wonder about the biomass curve at Ph 9, please explain it.
Three value of pH was selected as neutral (pH 7), light alkaline (pH 9) and alkaline (pH 10). Previously, we found that the strain is an alkalitolerant fungus and grew well in the pH range from neutral to alkaline up to pH 10,5 [doi:10.5598/imafungus.2013.04.02.07.]. A special alkaline growth medium has been developed previously with pH 10[doi:10.1134/S0003683817060035]. The best grows for the E101 strain was detected at pH 9.
There are a lot of spelling mistakes in the manuscript, such as abbreviations without full names, chemical formulas
We have corrected omissions and typos in the text, figure legends and additional files.
Reviewer 2 Report
The article entitled Uncovering the Effects of Cultivation Condition on Different Form of Peptaibol’s Emericellipsins Production from an Alkaliphilic Fungus Emericellopsis alkaline is a brilliant work done by the team. The work demonstrates the role of Peptaibol bioactivity in biocontrol of pathogen and its production in varying environmental conditions. The manuscript is worth reading and publication with potential to attract large number of readerships. However, a few concerns are being highlighted , they must be dealt before acceptance
· A thorough grammar and English language check is mandatory.
· Acronym, Paib, must be used across the manuscript after peptaibols has been used once. Further check must be done to find out the correct use of acronym
· Line 56: A dot can be removed before citation.
· Avoid unnecessary upper-case characters in the middle of the sentence for example:
Line 68 (Extreme Environments), 80 (Light microscope ), 81 (Scanning electron microscope (SEM)
· Line 78: please correct the expression of pH (pH 7, 9, 10 (±0.2)), I meant ±0.2 for each pH? How was it precisely ±0.2 for each value?
· Is it possible to provide error bars on Figure 1 and 2
1. language of the manuscript must be check correctly
2. grammar and typos must be checked
Author Response
The article entitled Uncovering the Effects of Cultivation Condition on Different Form of Peptaibol’s Emericellipsins Production from an Alkaliphilic Fungus Emericellopsis alkaline is a brilliant work done by the team. The work demonstrates the role of Peptaibol bioactivity in biocontrol of pathogen and its production in varying environmental conditions. The manuscript is worth reading and publication with potential to attract large number of readerships. However, a few concerns are being highlighted , they must be dealt before acceptance
Thank you very much for the evaluation of the manuscript. We are resubmitting our revised manuscript and detailed point-by-point responses. We appreciate your constructive comments!
- A thorough grammar and English language check is mandatory.
- Acronym, Paib, must be used across the manuscript after peptaibols has been used once. Further check must be done to find out the correct use of acronym
We thank the Reviewer for pointing this out. We have revised it within all text.
- Line 56: A dot can be removed before citation.
- Avoid unnecessary upper-case characters in the middle of the sentence for example:
Line 68 (Extreme Environments), 80 (Light microscope ), 81 (Scanning electron microscope (SEM)
We have corrected omissions and typos in the text, figure legends and additional files.
Line 78: please correct the expression of pH (pH 7, 9, 10 (±0.2)), I meant ±0.2 for each pH? How was it precisely ±0.2 for each value?
We have revised this sentence.
Is it possible to provide error bars on Figure 1 and 2
Thanks for this notification. We are used three replicates; we have added the error bars and revised the in Figure 2A and Figure 2B
Comments on the Quality of English Language
- language of the manuscript must be check correctly
- grammar and typos must be checked
We are grateful for revealed mistake, we had fixed it.
Reviewer 3 Report
Anastasia E. Kuvarina et al. have investigated the effects of various pH and cultivation condition on peptaibol complex Emericellipsins A-E (EmiA-E) produced by alkaliphilic fungus Emericellopsis Alkaline in this study. The overall experimental design concept of this article is acceptable, the manuscript is clear, and the structure is good. Here are some questions.
1. The third sentence in the introduction cites 10 references, isn't it too much? If the first three sentences cited these 10 references, is it more appropriate to cite them separately?
2. What is the reason for choosing EtOAc or butanol for extraction?
3. The experimental method and comparison with previous research results should not appear in the results section.
4. Is it sufficient to identify EmiA and only determine its quality through LC-MS? If the results of nuclear magnetic resonance identification were added, it would be better.
5. When exploring the three factors of pH, time, and culture conditions, have only one set of other conditions been established? In other words, when exploring the effect of pH on EmiA, what are its cultivation conditions? If both cultivation conditions are present, which set of data was used for the drawing? Why chooses that group?
6. The author's results indicate that alkaline medium significantly increased Paib production, while biomass production decreased. What is the reason for this phenomenon? Does Paib have an inhibitory effect on the Emericellopsis Alkalina strain itself?
A large number of hyphens appear in this article, such as"re-lates", "stru-ture", and so on, which makes the reading process not very comfortable.
Author Response
Anastasia E. Kuvarina et al. have investigated the effects of various pH and cultivation condition on peptaibol complex Emericellipsins A-E (EmiA-E) produced by alkaliphilic fungus Emericellopsis Alkaline in this study. The overall experimental design concept of this article is acceptable, the manuscript is clear, and the structure is good. Here are some questions.
Thank you very much for the evaluation of the manuscript. We are resubmitting our revised manuscript and detailed point-by-point responses. We appreciate your constructive comments!
- The third sentence in the introduction cites 10 references, isn't it too much? If the first three sentences cited these 10 references, is it more appropriate to cite them separately?
We divided the references according to the meaning.
- What is the reason for choosing EtOAc or butanol for extraction?
Liquid-liquid extraction with ethyl acetate was used, according to the methodology described early[doi:10.3390/molecules23112785]. Many studies show that ethyl acetate the appropriate solvent for peptaibols extraction.
- The experimental method and comparison with previous research results should not appear in the results section.
Thank you for your comments. We moved it to the discussion.
- Is it sufficient to identify EmiA and only determine its quality through LC-MS? If the results of nuclear magnetic resonance identification were added, it would be better.
Thank you for your comments. Therefore, we should inform you that still we were caring out our screenings at all possible cases we observed only one component with mass of 1050.7 Da and there was no one more compound with such mass. On the other hand, we have examined the fragmentation spectrum of EmiA component with MALDI-TOF Ms|Ms method application and masses of fragments and the character of fragmentation are in a good correlation with EmiA structure earlier obtained with NMR spectroscopy [Rogozhin et al.,2018]. doi:10.3390/molecules23112785
Taking in account all another EmiA “homologues”, we did not try to identify them as real homologues; we only marked them as potential peptaibols which must be taken in account for the next researches including individual separation from total mixture and its’ structural identification particularly with NMR spectroscopy application. Moreover, the current work could not be exhibited as any structural discovery, so for our purpose herein Ms method was more appropriate besides NMR.
- When exploring the three factors of pH, time, and culture conditions, have only one set of other conditions been established? In other words, when exploring the effect of pH on EmiA, what are its cultivation conditions? If both cultivation conditions are present, which set of data was used for the drawing? Why chooses that group?
Previously, we found that the strain is an alkalitolerant fungus and grew well in the pH range from neutral to alkaline[doi:10.5598/imafungus.2013.04.02.07.]. The present study deals with the influence of pH and the method of cultivation on the accumulation of peptaibols, both factors were studied simultaneously. The aim of the work was to evaluate the effect of pH from neutral to alkaline in different types of cultivation. The effect of each of the three pH levels was evaluated during growth under stationary and submerged conditions.
- The author's results indicate that alkaline medium significantly increased Paibproduction, while biomass production decreased. What is the reason for this phenomenon? Does Paib have an inhibitory effect on the Emericellopsis Alkalina strain itself?
Yeas, EmiA have slightly inhibitory effect on the producer in concentration upper to 300 µg/mL
Comments on the Quality of English Language
A large number of hyphens appear in this article, such as"re-lates", "stru-ture", and so on, which makes the reading process not very comfortable.
We are grateful for revealed mistake, we had fixed it.